# Potential Benefits of Photon-Counting CT in Dental Imaging: A Narrative Review

**DOI:** 10.3390/jcm13082436

**Published:** 2024-04-22

**Authors:** Chiara Zanon, Alessia Pepe, Filippo Cademartiri, Costanza Bini, Erica Maffei, Emilio Quaia, Edoardo Stellini, Adolfo Di Fiore

**Affiliations:** 1Department of Radiology, University of Padua, 35128 Padova, Italy; 2Department of Radiology, Fondazione Toscana Gabriele Monasterio, 56124 Pisa, Italy; 3IRCCS Synlab SDN, 80143 Naples, Italy; 4Department of Neuroscience, School of Dentistry, Division of Prosthodontics and Digital Dentistry, University of Padova, 35122 Padova, Italy

**Keywords:** photon-counting computed tomography, dental imaging, radiation exposure, image artifacts, cone-beam computed tomography, endodontic imaging, iterative metal artifact reduction, virtual monoenergetic imaging

## Abstract

**Background/Objectives:** Advancements in oral imaging technology are continually shaping the landscape of dental diagnosis and treatment planning. Among these, photon-counting computed tomography (PCCT), introduced in 2021, has emerged as a promising, high-quality oral technology. Dental imaging typically requires a resolution beyond the standard CT systems achievable with the specialized cone-beam CT. PCCT can offer up to 100 µm resolution, improve soft-tissue contrast, and provide faster scanning times, which are crucial for detailed dental diagnosis and treatment planning. Using semiconductor detectors, PCCT produces sharper images and can potentially reduce the number of scans required, thereby decreasing patient radiation exposure. This review aimed to explore the potential benefits of PCCT in dental imaging. **Methods:** This review analyzed the literature on PCCT in dental imaging from January 2010 to February 2024, sourced from PubMed, Scopus, and Web of Science databases, focusing on high-resolution, patient safety, and diagnostic efficiency in dental structure assessment. We included English-language articles, case studies, letters, observational studies, and randomized controlled trials while excluding duplicates and studies unrelated to PCCT’s application in dental imaging. **Results:** Studies have highlighted the superiority of PCCT in reducing artifacts, which are often problematic, compared to conventional CBCT and traditional CT scans, due to metallic dental implants, particularly when used with virtual monoenergetic imaging and iterative metal artifact reduction, thereby improving implant imaging. This review acknowledges limitations, such as the potential for overlooking other advanced imaging technologies, a narrow study timeframe, the lack of real-world clinical application data in this field, and costs. **Conclusions:** PCCT represents a promising advancement in dental imaging, offering high-resolution visuals, enhanced contrast, and rapid scanning with reduced radiation exposure.

## 1. Introduction

The evolution of CT technology has paralleled advancements in computer science, X-ray production, and detection, culminating in novel applications, such as dual-energy CT (DECT) for medical imaging [1]. A significant leap has been made with the introduction of energy-resolving photon-counting detectors (PCDs), which are distinguished by their ability to discern the energy of individual X-ray photons, offering improved spatial resolution, noise reduction, dose efficiency, and spectral imaging [1]. The introduction of clinical photon-counting CT (PCCT) systems in 2021 has led to significant advancements in CT technology, offering the potential for lower patient doses and higher spatial resolution owing to smaller detector pixels [2,3,4,5,6]. This innovation is poised to update clinical applications, particularly in anatomical imaging, by providing clearer images of dental structures (5). PCCT promises to provide high-resolution images (up to 100 µm), superior soft-tissue contrast, and faster scanning times, which are crucial for detailed dental diagnoses and treatment planning (Figure 1 and Figure 2).

Metal implants in traditional CT scans often create artifacts that can lead to misdiagnosis owing to obscured images resulting from photon starvation, scattering, and beam hardening [7]. PCCT introduces new artifact reduction methods that enhance diagnostic imaging in patients with dental and orthopedic implants [8]. These techniques include Iterative Metal Artifact Reduction (iMAR), which methodically reduces artifacts; high kilo-electron volt (keV) virtual monoenergetic imaging (VMI) for clearer images; and sinogram inpainting to improve image quality around implants [9]. These advances allow for the more precise visualization of dental structures and decrease the prevalence of misleading artifacts (Figure 3).

This review evaluates the efficacy, safety, and precision of image quality in PCCT, comparing it to the current standard of cone-beam CT (CBCT). It examines research from 2021 to 2023, focusing on the potential of PCCT to improve diagnostic efficiency in dental structure assessment and reduce artifacts more effectively than other imaging modalities.

This review examines PCCT’s applications, challenges, and prospective trajectory in dental diagnostics. Key inquiries addressed include the comparison of spatial resolution and radiation dosage between ultra-high-resolution PCCT and CBCT, particularly for detecting pathologies such as apical osteolysis [10]. Furthermore, this review explored whether PCCT can outperform microCT in visualizing complex endodontic structures and reducing metal artifacts from dental implants.

MicroCT, CBCT, and PCCT, are imaging techniques based on X-rays [11,12]. CBCT uses a cone-shaped source of ionizing radiation and a two-dimensional detector that provides multidimensional images. It is primarily used for planning dental implants and surgical tooth extraction. However, the regular prescription of CBCT in all patients increases the collective radiation dose for orthodontic patients [11,13].

The main principle of micro-CT is the generation of a series of radiographs of a sample placed between an X-ray source and detector. The mode of action is similar to that of CT scans. The main difference lies in the smaller size of the focal-spot X-ray tube, which provides higher-resolution images. Although dense structures such as bones can be visualized without any specific preparation, a limitation of micro-CT scanners is the low contrast of soft tissues due to low X-ray absorption [12,14].

Additionally, we assessed the effectiveness of VMI and iMAR techniques in enhancing image quality and reducing artifacts within PCCT frameworks [15,16].

## 2. Photon-Counting Detector—Technical Considerations

PCCT uses PCDs to enhance medical imaging by directly measuring the energy of X-ray photons as they pass through the patient [2,3,4,5,6]. Unlike conventional CT scanners that use energy-integrating detectors (EIDs), which only indirectly measure energy, PCDs absorb photons and convert them to electrical signals [6]. This allows for the precise counting and energy differentiation of photons, which improves image quality by increasing spatial and contrast resolution while reducing noise, artifacts, and patient radiation exposure [5].

Additionally, PCCT is capable of multi-energy imaging, as it distinguishes between different tissue types and contrast agents based on their atomic properties [6].

PCDs transform incoming photons into electron–hole pairs, resulting in an electrical signal that is proportional to the photon energy. This ability minimizes electronic noise, excluding low-energy photons, and avoids beam-hardening artifacts around dense structures, such as bones. Simultaneous multi-energy imaging allows the optimization of energy thresholds for various diagnostic applications, including K-edge imaging [2,3,4,5,6].

Three-dimensional (3D) image reconstructions using the volume rendering technique (VRT) are helpful for advanced analysis in clinical settings, for enhancing visualization and interpretation, and for understanding complex anatomical and dental structures. They are beneficial for diagnosis, the treatment planning of maxillofacial lesions, educational demonstrations, and teaching in medical training [17] (Figure 4 and Figure 5).

## 3. Methodology

An electronic search was performed on PubMed, Scopus, and Web of Science databases to identify relevant publications concerning PCCT in dental imaging, encompassing articles published from January 2010 to February 2024.

The search strategy used was as follows: “((photon counting CT))” and “((Dental))”. Articles published in English were included. The final review excluded duplicate studies and articles that did not specifically address the role of photon-counting CT in dental imaging. Evidence from various study types, including case studies, letters, observational studies, and randomized controlled trials, was examined.

## 4. Results

The characteristics of the selected studies are summarized in Table 1.

## 5. PCCT versus CBCT in Endodontic Imaging

Ruetters et al. [10] assessed PCCT performance with a spatial resolution of approximately 200 µm compared to conventional dental CBCT. Both modalities identified all eight cases of apical osteolysis. Significant differences favored PCCT at medium and high doses for root canals (*p* = 0.0001), periodontal space (*p* = 0.0090), cortical (*p* = 0.0003), and spongious bone (*p* = 0.0293). Excellent image quality was reported for both devices across all structures. The study used five cadaver heads, scanned with an ultra-high-resolution PCCT (pixel size 0.25 mm at isocenter) and CBCT, at doses of 8.5 mGy to 66.5 mGy for PCCT and 8.94 mGy for CBCT. It was concluded that ultra-high-resolution PCCT could reliably identify dental pathologies with a radiation dose similar to that of conventional CBCT.

Fontenele et al. [18] compared PCCT and CBCT for endodontic imaging precision, using an anthropomorphic phantom aligned with industrial micro-CT as a reference. The phantom was scanned using a PCCT (NAEOTOM Alpha) and two CBCTs (high-resolution 3D Accuitomo 170 and NewTom VGi evo) at various resolutions. This study aimed to accurately depict complex endodontic structures, such as apical deltas, narrow canals, isthmuses, and root cracks. Evaluations by three expert examiners revealed that PCCT paralleled the reference in terms of structural detail (*p* > 0.05).

In essence, both PCCT and high-resolution 3D Accuitomo 170 CBCTs excel in visualizing fine structures compared to the industrial microCT scanner; additionally, the high-resolution 3D Accuitomo 170 is superior in detecting root cracks (*p* < 0.05).

Vanden Broeke et al. [19] evaluated the use of PCCT, focusing on detecting accessory canals (ACs) and minimizing metal artifacts from dental implants. The study involved scanning eight extracted teeth, six with ACs, one with a titanium rod, and another with a gutta-percha point, using spectral PCCT, CBCT, and microtomography (microCT). Spectral PCCT and CBCT were equally effective for visualizing ACs. However, spectral PCCT and microCT showed comparable efficacy in reducing metal artifacts, surpassing that of CBCT. While microCT delivers high-resolution images of minute anatomical structures, its use is limited to nonclinical settings. Spectral PCCT, equaling CBCT’s detection capabilities and excelling artifact reduction, shows promising potential for advanced dental imaging, combining the practicality of CBCT with the high detail of microCT.

## 6. PCCT in Metal Artifact Reduction and Dose Reduction

Anhaus et al. [20] refined the metal artifact reduction techniques for clinical PCCT. By testing with phantoms designed to mimic various body regions (hip, teeth, spine, nerves) and using a Gammex phantom for precision in Hounsfield Unit (HU) accuracy, scans were performed at 120 kV and tin-filtered Sn140 kV. They focused on determining the most effective settings for VMI and iterative Metal Artifact Reduction (iMAR). Their findings identified 100 keV and 110 keV as the optimal VMI energy levels for the 120 kV and Sn140 kV scans, respectively, for the most accurate HU restoration. This study conclusively demonstrated that iMAR significantly alleviates metal artifacts, emphasizing the ability of PCCT to fine-tune artifact reduction.

Pallasch et al. [15] aimed to mitigate metal artifacts in the CT scans of patients with dental implants using PCCT. The study included 48 patients with a mean CT dose index volume (CTDIvol) of 6.2 mGy. They compared standard PCCT, PCCT with monoenergetic 140 keV, PCCT with the iterative Metal Artifact Reduction (iMAR) algorithm, and a combination of iMAR with 140 keV imaging. Radiologist evaluations and quantitative analyses showed that iMAR significantly improved image quality and reduced artifacts (*p* < 0.001) without affecting the coefficient of variation. Notably, iMAR was more effective than the monoenergetic reconstruction at 140 keV. The study affirmed that PCCT combined with iMAR is a potent tool for enhancing diagnostics in patients with dental hardware, potentially uncovering pathologies previously concealed by metal artifacts without increasing radiation exposure.

Patzer et al. [16] explored the use of VMI and iterative Metal Artifact Reduction (iMAR) in PCCT to enhance the images of dental implants. The study involved 50 patients and assessed the performance of traditional polychromatic 120 kVp imaging against VMI and iMAR techniques. They measured artifact attenuation and noise using three readers evaluating image quality and artifact presence. This study established that iMAR significantly reduced artifacts and improved tissue clarity. VMI combined with iMAR at energies of 110 keV or higher outperformed traditional methods in artifact reduction, whereas VMI alone was less effective. High-energy VMI minimized soft tissue damage, and the dual VMI and iMAR methods reduced overcorrection artifacts. The research concluded that combining VMI with iMAR greatly enhances PCCT imaging for patients with dental implants, offering a substantial improvement in diagnostic quality and artifact management in maxillofacial imaging.

Risch et al. [21] conducted a focused evaluation of diminishing dental material artifacts in the PCCT datasets. Recruiting a cohort of 50 patients requiring neck CT scans, this study compared the artifact reduction capabilities of standard and sharp kernels, with and without iMAR, across VMI levels ranging from 40 keV to 190 keV. Artifact severity was quantitatively assessed using the mean absolute error and artifact index (AIX) across various slices. The results indicated a substantial artifact reduction for VMI levels above 70 keV when using iMAR, with a peak decrease of 25%. However, the sharp kernel, especially when paired with iMAR, increased AIX by up to 38%. The most pronounced improvement was observed with iMAR alone, where the mean absolute error decreased by up to 84% and AIX by up to 90%. The study concluded that iMAR robustly reduces metal artifacts in dental CT imaging, surpassing the variations in kernel or VMI settings, and that combining higher VMI levels with iMAR yields further enhancement.

Layer et al.’s research [22] assessed the efficacy of VMI for reducing artifacts from dental implants in PCCT. By analyzing 30 PCCT scans with significant dental implant artifacts, images were reconstructed using VMI at energy levels ranging from 100 keV to 190 keV. Subjective evaluations and objective measures across different Regions of Interest (ROIs) revealed substantial improvements in image quality at all VMI levels compared with standard images, particularly at 130 keV. This level showed the most significant artifact attenuation, which enhanced the clarity of the soft palate and cheeks. Quantitative analysis confirmed that 130 keV was the optimal VMI energy, effectively aligning tissue attenuation values with artifact-free regions. The study concluded that VMI, especially at 130 keV, significantly mitigates implant-related artifacts in PCCT, providing clearer imaging of the head and neck, and improving diagnostic accuracy.

Sawall et al. [23] evaluated the image quality between a low-dose PCCT protocol and a normal-dose Digital Volume Tomography (DVT) system. In this study, ten porcine jaws were scanned using both PCCT (90 kV, 1 mGy CTDI16 cm) and DVT (85 kV, 4 mGy) with matched resolutions. Images were reconstructed using a voxel size of 160 × 160 × 200 µm. The dose-normalized contrast-to-noise ratio (CNRD) for the dentin enamel and dentin bone was then calculated. The findings revealed that PCCT achieved a 37% higher CNRD for dentin-enamel and 31% higher CNRD for dentin-bone contrast than DVT, with both contrasts being statistically significant (*p* < 0.05). Diagnostic image quality received higher ratings for PCCT, preferred by both readers over DVT (*p* < 0.02 and *p* < 0.04, respectively). The visibility of anatomical structures also scored higher on PCCT images (all *p* < 0.05), and reader reproducibility was acceptable (Intraclass Correlation Coefficient > 0.64). The study concluded that PCCT provides superior image quality compared to DVT at a lower radiation dose, indicating its potential for advanced dental imaging applications with reduced patient exposure.

The study by Lee et al. [24] introduced a novel triple-energy virtual monochromatic imaging (TEVM) technique, utilizing PCD technology, to improve dental computed tomography (DCT) image quality when marred by metal artifacts. By synthesizing images from three distinct energy levels to create a VMI, TEVM enhances contrast and sharpness, particularly in areas affected by metal restorations or implants. The efficacy of this technique for artifact reduction and image enhancement was evaluated against traditional imaging methods in dental and endodontic CT scans. Results showed that TEVM significantly reduces metal artifacts, potentially reducing the need for multiple scans and lowering radiation exposure for patients. TEVM with PCD is highlighted as a significant advancement for increasing diagnostic accuracy and improving patient care in the dental field, offering a solution to a longstanding problem in dental imaging.

## 7. Discussion

PCCT employs PCDs to directly measure X-ray photon energy, enhance image resolution and clarity, minimize exposure and artifacts, and facilitate multi-energy imaging for accurate diagnostics [2,3,4]. PCCT can offer resolutions of up to 100 µm, enhancing soft-tissue contrast and providing quicker scans, contributing to patient safety through reduced radiation. Utilizing semiconductor detectors, PCCT has significantly reduced artifact interference, especially from metals in implants, with the aid of iterative iMAR and VMI [2,3,4].

In endodontic imaging, PCCT has been compared to conventional CBCT. Ruetters et al. found that PCCT, at a radiation dose similar to CBCT, provided superior resolution for detecting apical osteolysis and detailed dental structures [10]. Vanden Broeke et al. demonstrated that spectral PCCT was as effective as CBCT in visualizing accessory canals but superior in reducing metal artifacts, suggesting that PCCT merges the practical aspects of CBCT with detailed imaging [19]. These findings indicate PCCT’s potential to enhance dental diagnostics, owing to its high-resolution capabilities and efficient artifact reduction.

Additionally, Anhaus et al. identified optimal VMI settings, with iMAR proving to be significantly effective for artifact reduction [21]. Pallasch et al. [15] demonstrated that iMAR alone, especially in combination with 140 keV VMI, greatly enhances image quality and clarity in patients with dental hardware, without increasing radiation exposure. Similarly, Patzer et al. found that iMAR, especially when used with higher-energy VMI, outperformed traditional imaging methods in reducing artifacts and improving tissue clarity in dental implant images [16]. Layer et al. highlighted the efficacy of VMI at 130 keV in substantially improving the clarity of images with dental implant artifacts [22]. Sawall et al. confirmed that PCCT provides superior image quality compared to Digital Volume Tomography (DVT) at lower radiation doses, indicating its potential for advanced dental imaging applications with reduced patient exposure [23].

These studies collectively advocate for the advanced capabilities of PCCT in dental imaging, particularly artifact management and dose optimization. These images can be used by dental professionals for diagnostic purposes, treatment planning, and to track the progress of treatment, particularly in orthodontics, implantology, and other oral surgeries. However, challenges persist, such as in overlooking other imaging technologies, the recent introduction of PCCT that limits long-term study data, the scarcity of real-world applications, and the cost of this new technology. Further research is essential to realize the full potential of PCCT in the clinical setting, expand its use in the field, and establish its cost-effectiveness in comparison to traditional imaging methods such as CBCT.

## 8. Limitations

This review highlights PCCT advancements in dental diagnostics; however, it has the following limitations.

PCCT provides enhanced spatial resolution compared to traditional CT scans; however, its application in dental imaging is limited. The absence of specialized software and standardized processing protocols restrict its clinical utility.

While PCCT represents a significant step in enhancing dental imaging quality and patient safety, the evaluation within the 2021–2023 period has several limitations. The restricted study timeline leads to a potential underrepresentation of long-term data and technological evolution.

Moreover, the real-world application of PCCT has not been sufficiently documented, which raises questions regarding its practicality and efficiency in everyday clinical settings.

Additionally, the financial implications of adopting PCCT technology have not been fully explored in this review. The high cost associated with PCCT systems, which require multiple generators and X-ray tubes, affects their affordability and adoption. The high costs and the limited availability of PCCT could limit the application of this technology in dental imaging, potentially impacting the equitable provision of cutting-edge dental care. This limitation underscores the pressing need for cost-effectiveness studies and broader economic analysis. Furthermore, these studies predominantly involved non-live patient samples, casting doubt on the generalizability of the findings.

Comparative analyses using conventional CBCT are limited and may not represent a full range of clinical scenarios.

The focus on artifact reduction might also overshadow the necessity of comprehensively investigating the diagnostic accuracy. This review emphasizes artifact reduction, without considering the possibility of diagnostic oversight. It also lacks discussions on statistical validation methods and regulatory and ethical considerations and omits patient-centered outcomes. These limitations underscore the necessity for further research to validate PCCT’s clinical and economic value of PCCT in dental diagnostics.

Thus, future research should address these gaps, providing a more comprehensive and long-term perspective on the clinical efficacy and practicality of PCCT, ensuring that it meets the pragmatic demands of dental healthcare delivery. Ongoing research aimed at refining PCCT protocols and enhancing postprocessing methods is crucial for advancing the diagnostic capabilities of dental imaging.

## 9. Conclusions

In conclusion, PCCT can represent a development in dental imaging, offering high-resolution visuals, improved contrast, and rapid scanning capabilities, while minimizing radiation exposure. Studies from 2021 to 2023 have demonstrated PCCT’s superior performance in endodontic imaging and artifact reduction compared to conventional CBCT, particularly in visualizing fine structures and reducing artifacts from dental hardware. The integration of VMI and iMAR further enhances PCCT’s diagnostic precision. Although this review has carefully evaluated PCCT against existing imaging modalities, it acknowledges limitations such as the narrow timeframe of studies, potential biases, and a focus on artifact reduction that might overshadow other diagnostic aspects. These insights pave the way for future research to optimize PCCT’s clinical application of PCCT, ensuring that it provides support in dental practice with proven efficacy, cost-effectiveness, and patient-centric outcomes.

In conclusion, PCCT can provide images with a high resolution of up to 100 µm, high soft-tissue contrast, and rapid scanning, thus enhancing diagnostic capabilities while reducing radiation exposure. Studies conducted between 2021 and 2023 demonstrated PCCT’s effectiveness in endodontic imaging and metal artifact reduction compared with CBCT. Technologies such as VMI and iMAR have shown promise in optimizing PCCT’s visualization of dental structures and pathologies of PCCTs, indicating their potential in dental imaging. However, it is important to note that the conclusion of this study should be approached with caution due to its significant limitations.

## Figures and Tables

**Figure 1 jcm-13-02436-f001:**
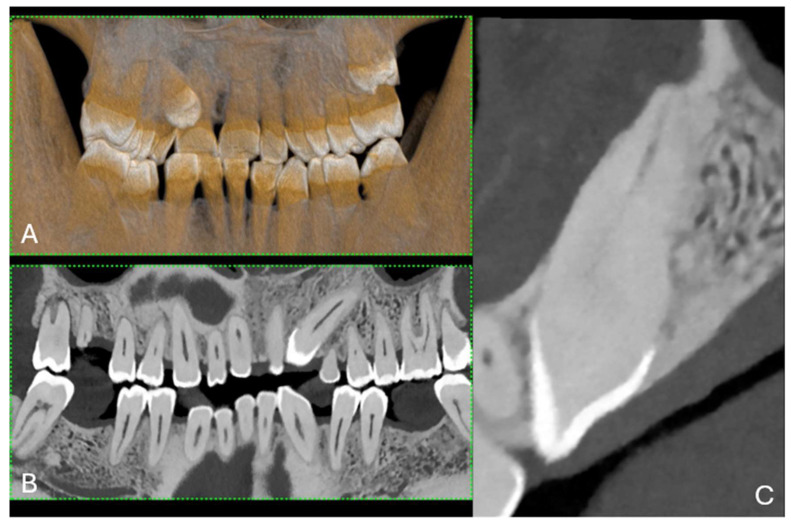
(**A**) A 3D cinematic rendering of the occlusal view of the lower jaw, where the crowns of molars and premolars are visible. (**B**) Longitudinal MPR and axial cross-section of upper and lower jaws. (**C**) The region of a single tooth and root structure in relation to the jawbone. The scan was performed on a commercial whole-body Dual Source Photon Counting CT scanner (NAEOTOM Alpha, Siemens Healthineers, Erlangen, Germany); 0.2 mm slice thickness, 0.1 mm reconstruction increment, FOV 140 mm, spiral acquisition with tube current modulation; resolution matrix of 1024 × 1024 pixels on the source axial reconstructions with a kernel filtering of Bv72; maximum intensity of Quantum Iterative Reconstruction (QIR 4). The actual displayed resolution is 0.1 mm (100 microns). Abbreviations: 3D, three-dimensional; MPR, multiplanar reconstruction; FOV, field-of-view.

**Figure 2 jcm-13-02436-f002:**
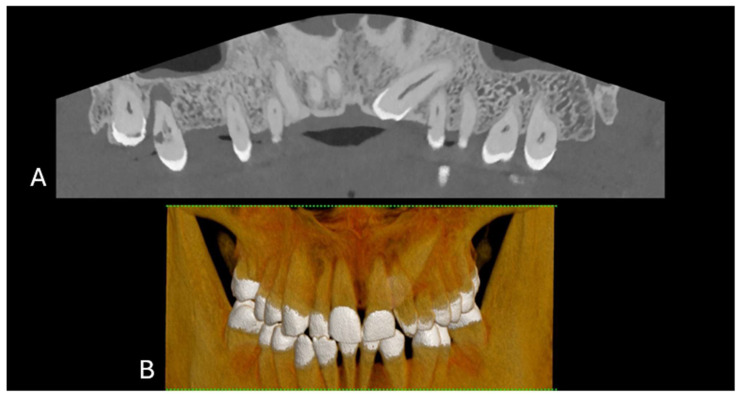
(**A**) The curved MPR of the upper and lower jaws, the roots of the teeth, and the surrounding bone structure, which is commonly used to assess overall dental health, including bone levels and root configurations, and to check for any abnormalities such as impacted teeth or pathologies in the jawbone. (**B**) Front-view 3D cinematic rendering of the jaws and teeth, which shows the relative positions of the teeth in both the upper and lower jaws. This type of visualization helps to assess the bite, spacing, and alignment of the teeth, as well as the relationship between the upper and lower dental arches. The scan was performed on a commercial whole-body Dual Source Photon Counting CT scanner (NAEOTOM Alpha, Siemens Healthineers); 0.2 mm slice thickness, 0.1 mm reconstruction increment, FOV 140 mm, spiral acquisition with tube current modulation; resolution matrix of 1024 × 1024 pixels on the source axial reconstructions with a kernel filtering of Bv72; maximum intensity of Quantum Iterative Reconstruction (QIR 4). The actual displayed resolution is 0.1 mm (100 microns). Abbreviations: 3D, three-dimensional; MPR, multiplanar reconstruction; FOV, field-of-view.

**Figure 3 jcm-13-02436-f003:**
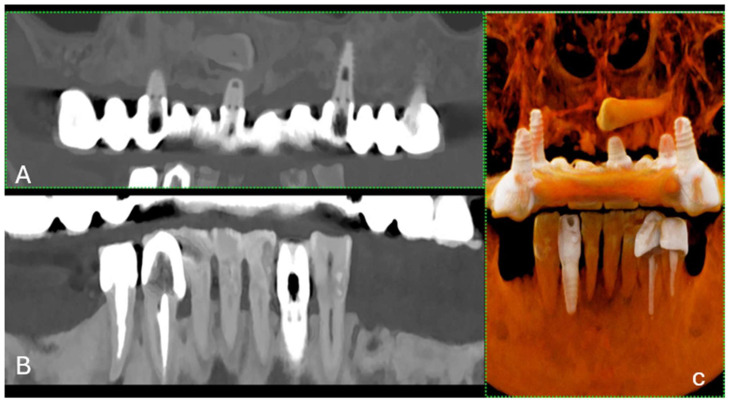
(**A**) A longitudinal MPR (A) and panoramic view of the upper and lower jaws, displaying a metallic dental prosthesis. Unlike conventional CT, there are no metal artifacts. (**B**) A curved MPR image of a portion of the jaws, highlighting a detailed view of several teeth and their respective roots. (**C**) The 3D cinematic rendering, providing a realistic perspective of the teeth’s condition and alignment, including the dental implant. The scan was performed on a commercial whole-body Dual Source Photon Counting CT scanner (NAEOTOM Alpha, Siemens Healthineers); 0.2 mm slice thickness, 0.1 mm reconstruction increment, FOV 140 mm, spiral acquisition with tube current modulation; resolution matrix of 1024 × 1024 pixels on the source axial reconstructions VIM at 190 KeV. The actual displayed resolution is 0.1 mm (100 microns). Abbreviations: 3D, Three-Dimensional; VMI = Virtual Monoenergetic Imaging; FOV = Field of View.

**Figure 4 jcm-13-02436-f004:**
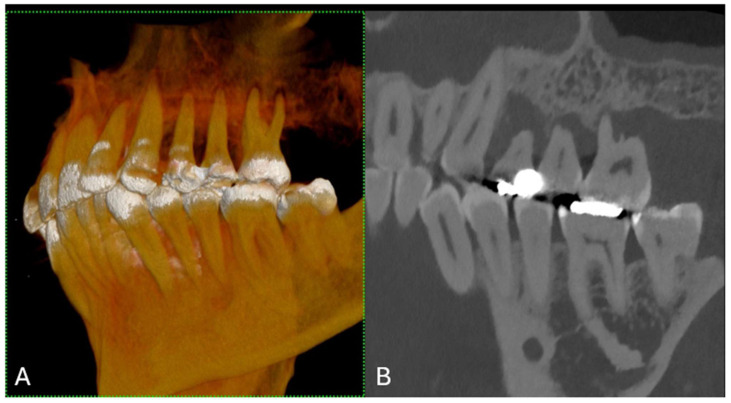
(**A**) A 3D cinematic rendering that displays the detailed structure of a jaw with teeth, emphasizing their three-dimensional form and alignment. (**B**) The longitudinal MPR showing the teeth, fillings, and jawbone with high contrast, which is useful for evaluating dental health and anatomy. The scan was performed on a commercial whole-body Dual Source Photon Counting CT scanner (NAEOTOM Alpha, Siemens Healthineers); 0.2 mm slice thickness, 0.1 mm reconstruction increment, FOV 140 mm, spiral acquisition with tube current modulation; resolution matrix of 1024 × 1024 pixels on the source axial reconstructions with a kernel filtering of Bv72; maximum intensity of Quantum Iterative Reconstruction (QIR 4). The actual displayed resolution is 0.1 mm (100 microns). Abbreviations: 3D, three-dimensional; MPR, multiplanar reconstruction; FOV, field-of-view.

**Figure 5 jcm-13-02436-f005:**
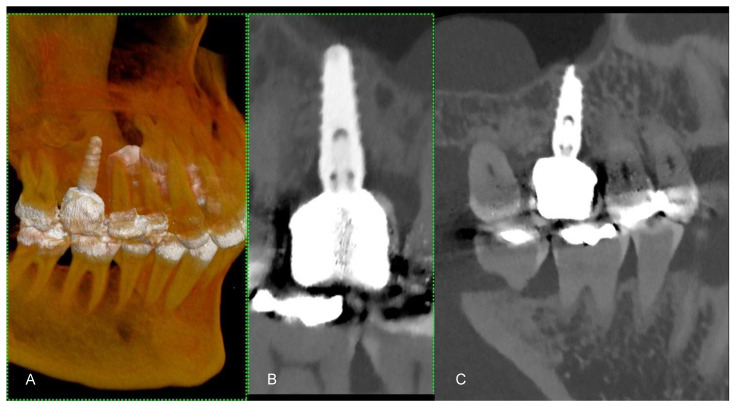
(**A**) The figure shows a 3D cinematic rendering of dental implants. This type of visualization helps to ensure that the implants are properly positioned for the best functional and aesthetic outcomes. (**B**) The 2D cross-sectional image of a single dental implant in the jawbone. This type of image is useful for evaluating the quality and density of the bone surrounding the implant and ensuring that the implant is properly placed. (**C**) This is another cross-sectional study, which also shows the detailed interaction between the implants and the bone, ensuring that the implants are not encroaching upon any anatomical structures, such as nerves or sinuses, and that there is enough bone around the implants for a stable fit. The scan was performed on a commercial whole-body Dual Source Photon Counting CT scanner (NAEOTOM Alpha, Siemens Healthineers); 0.2 mm slice thickness, 0.1 mm reconstruction increment, FOV 140 mm, spiral acquisition with tube current modulation; resolution matrix of 1024 × 1024 pixels on the source axial reconstructions VIM at 190 KeV. The actual displayed resolution is 0.1 mm (100 microns). Abbreviations: 3D, Three-Dimensional; VMI = Virtual Monoenergetic Imaging; FOV = Field of View.

**Table 1 jcm-13-02436-t001:** Photon-counting computed tomography in dental imaging studies.

		PCCT	CBCT	
PCCT vs. CBCT in Endodontic Imaging	Ruetters et al. *Sci Rep* 2022 [10]	+		Ability to detect apical osteolysis and dental structures.
	Fontenele et al. *Sci Rep* 2023 [18]		+	Detecting root cracks.
	Vanden Broeke et al. *BDJ Open* 2021 [19]	+ + +	+	Detecting accessory canals. Reducing metal artifacts. Artifact reduction.
PCCT in Metal Artifact Reduction and Dose Reduction	Anhaus et al. *Phys Med Biol*. 2022 [20]	Anhaus et al. optimized PCCT metal artifact reduction with VMI and iMAR, finding optimal HU restoration at 100 keV (120 kV) and 110 keV (Sn140 kV), significantly mitigating artifacts in phantoms.		
	Pallasch et al. *Eur Radiol* 2023 [15]	The iMAR algorithm significantly enhanced image quality and reduced artifacts in CT scans with dental hardware compared to other methods. iMAR proved superior in maintaining signal homogeneity and improving diagnostic capabilities without increasing radiation exposure, over monoenergetic reconstructions at 140 keV.		
	Patzer et al. *Eur Radiol* 2023 [16]	iMAR significantly reduced dental implant artifacts and improved soft tissue clarity in PCCT imaging, with combined VMI and iMAR techniques outperforming traditional imaging.		
	Risch et al. *Invest Radiol* 2023 [21]	iMAR significantly lowers dental artifacts in PCCT, enhancing image quality. Higher VMI levels paired with iMAR provided the most substantial artifact reduction, affirming IMAR’s efficacy regardless of kernel choice.		
	Layer et al. *Sci Rep* 2024 [22]	VMI at 130 keV significantly reduced dental implant artifacts in PCCT scans compared to standard polychromatic images, improving soft tissue clarity.		
	Sawall et al. *J Dent* 2024 [23]	PCCT demonstrated superior image quality to DVT, achieving 37% higher contrast-to-noise ratio for dentin enamel and 31% for dentin bone at lower radiation doses. The study suggests PCCT’s potential for complex dental imaging with minimal radiation, providing high diagnostic accuracy and high score by readers for image quality preferred by readers.		
	Lee et al. *JINST* 2023 [24]	TEVM, which uses PCD technology to synthesize images from three energy levels, enhances DCT images affected by metal artifacts, improving contrast and sharpness. TEVM significantly outperforms traditional methods in reducing metal artifacts in dental and endodontic CT scans, minimizing the need for repeat scans and reducing radiation exposure.		

## Data Availability

Data are contained within the article.

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
