# Peer review of "Potential Benefits of Photon-Counting CT in Dental Imaging: A Narrative Review"

_jcm, 2024, doi:10.3390/jcm13082436_

Round 1

Reviewer 1 Report

Comments and Suggestions for Authors

The manuscript titled "Photon-Counting CT in Dental Imaging: Minimizing Exposure, Maximizing Detail" has been submitted to the Journal of Clinical Medicine.

While the manuscript addresses a compelling issue, there are several concerns regarding the study. Primarily, this systematic review does not comply with the elements indicated by the PRISMA guideline. In case of submitting a possible revised version, it is necessary to attach this guideline.

Title: The title is not informative. The purpose of the review should be redefined, perhaps to evaluate efficacy, minimize exposure, or maximize detail.

Abstract

It lacks the fundamental elements that this type of study should have. Please review what the PRISMA guideline indicates regarding this matter:

1. Provide an explicit statement of the main objective(s) or question(s) the review addresses.

2. Specify the information sources (e.g. databases, registers) used to identify studies and the date when each was last searched.

3. Specify the methods used to assess the risk of bias in the included studies.

4. Give the total number of included studies and participants and summarise relevant characteristics of studies.

5. Present results for main outcomes, preferably indicating the number of included studies and participants for each.

6. Provide a general interpretation of the results and important implications.

Keywords: Please ensure that all of them correspond to MeSH terms.

JCM does not require the presentation of key points.

Introduction

The problem and justification of the study are presented in a superficial manner. Moreover, it lacks the fundamental elements that this type of study should have.

Please review what the PRISMA guideline indicates regarding this matter:

1. Describe the rationale for the review in the context of existing knowledge.

2. Provide an explicit statement of the objective(s) or question(s) the review addresses.

It is not appropriate to present the questions in the introduction; that is the role of the methodology section. Furthermore, the questions should be consistent with the methodology, results, discussion, and conclusions, aspects that this review does not fulfill.

Methods

It lacks the fundamental elements that this type of study should have. Please review what the PRISMA guideline indicates regarding this matter:

1. Specify the inclusion and exclusion criteria for the review and how studies were grouped for the syntheses.

2. Specify all databases, registers, websites, organizations, reference lists and other sources searched or consulted to identify studies.

3. Specify the date when each source was last searched or consulted.

4. Present the full search strategies for all databases, registers, and websites, including any filters and limits used.

5. Specify the methods used to decide whether a study met the inclusion criteria of the review, including how many reviewers screened each record and each report retrieved, whether they worked independently, and if applicable, details of automation tools used in the process.

6. Specify the methods used to collect data from reports, including how many reviewers collected data from each report, whether they worked independently, any processes for obtaining or confirming data from study investigators, and if applicable, details of automation tools used in the process.

7. List and define all outcomes for which data were sought. Specify whether all results that were compatible with each outcome domain in each study were sought (e.g. for all measures, time points, analyses), and if not, the methods used to decide which results to collect.

8. List and define all other variables for which data were sought (e.g. participant and intervention characteristics, funding sources). Describe any assumptions made about any missing or unclear information.

9. Specify the methods used to assess risk of bias in the included studies, including details of the tool(s) used, how many reviewers assessed each study and whether they worked independently, and if applicable, details of automation tools used in the process.

10. Specify for each outcome the effect measure(s) (e.g. risk ratio, mean difference) used in the synthesis or presentation of results.

11. Describe the processes used to decide which studies were eligible for each synthesis (e.g. tabulating the study intervention characteristics and comparing against the planned groups for each synthesis.

12. Describe any methods required to prepare the data for presentation or synthesis, such as handling of missing summary statistics, or data conversions.

13. Describe any methods used to tabulate or visually display results of individual studies and syntheses.

14. Describe any methods used to synthesize results and provide a rationale for the choice(s).

15. Describe any methods used to assess risk of bias due to missing results in a synthesis (arising from reporting biases).

16. Describe any methods used to assess certainty (or confidence) in the body of evidence for an outcome.

The figure and Table 1 presented by the authors should be included in the results section.

All the information presented in the subheadings highlighted in bold should be included in the results section as indicated in that section.

Results

It lacks the fundamental elements that this type of study should have. Please review what the PRISMA guideline indicates regarding this matter:

1. Describe the results of the search and selection process, from the number of records identified in the search to the number of studies included in the review, ideally using a flow diagram.

2. Cite studies that might appear to meet the inclusion criteria, but which were excluded, and explain why they were excluded.

3. Cite each included study and present its characteristics.

4. Present assessments of risk of bias for each included study.

5. For all outcomes, present, for each study: (a) summary statistics for each group (where appropriate) and (b) an effect estimate and its precision (e.g. confidence/credible interval), ideally using structured tables or plots.

6. For each synthesis, briefly summarise the characteristics and risk of bias among contributing studies.

7. Present results of all statistical syntheses conducted.

8. Present results of all investigations of possible causes of heterogeneity among study results.

9. Present assessments of risk of bias due to missing results (arising from reporting biases) for each synthesis assessed.

10. Present assessments of certainty (or confidence) in the body of evidence for each outcome assessed.

Discussion

- The authors should reframe the discussion by initiating their arguments based on the studied objectives. Moreover, this review lacks the fundamental elements that this type of study should have.

- This study has many more limitations and biases that were not acknowledged.

Conclusions

It should be emphasized that the conclusions are based on significant limitations inherent in this study. This study has additional limitations that were not described. Considering the limitations present in the study, the authors draw very decisive conclusions. It is recommended to exercise caution in this regard. The results do not account for these conclusions.

The manuscript needs to be organized better; its presentation leaves much to be desired. Additionally, its language and grammar need to be reviewed and edited by the special services offered by the journal or other journals.

Also, review the referencing style as indicated in the author's instructions, as well as the parameters established by the journal.

Comments on the Quality of English Language

extensive editing is required

Reviewer 2 Report

Comments and Suggestions for Authors

Some Refs are missing

Irrelevant data exists

(The Authors must see my remarks)

Round 2

Reviewer 1 Report

Comments and Suggestions for Authors

Considering that the authors have decided to change the study design to a narrative review with a low level of evidence, the manuscript may be good.